# Shaping Social Activity by Incentivizing Users

**Mehrdad Farajtabar**[*]      **Nan Du**[*]      **Manuel Gomez-Rodriguez**[†]
**Isabel Valera**[‡]      **Hongyuan Zha**[*]      **Le Song**[*]
Georgia Institute of Technology[*]      MPI for Software Systems[†]      Univ. Carlos III in Madrid[‡]
{mehrdad,dunan}@gatech.edu      manuelgr@mpi-sws.org
{zha,lsong}@cc.gatech.edu      ivalera@tsc.uc3m.es

## Abstract

Events in an online social network can be categorized roughly into *endogenous* events, where users just respond to the actions of their neighbors within the network, or *exogenous* events, where users take actions due to drives external to the network. How much external drive should be provided to each user, such that the network activity can be steered towards a target state? In this paper, we model social events using multivariate Hawkes processes, which can capture both endogenous and exogenous event intensities, and derive a time dependent linear relation between the intensity of exogenous events and the overall network activity. Exploiting this connection, we develop a convex optimization framework for determining the required level of external drive in order for the network to reach a desired activity level. We experimented with event data gathered from Twitter, and show that our method can steer the activity of the network more accurately than alternatives.

## 1   Introduction

Online social platforms routinely track and record a large volume of event data, which may correspond to the usage of a service (*e.g.*, url shortening service, bit.ly). These events can be categorized roughly into *endogenous* events, where users just respond to the actions of their neighbors within the network, or *exogenous* events, where users take actions due to drives external to the network. For instance, a user's tweets may contain links provided by bit.ly, either due to his forwarding of a link from his friends, or due to his own initiative to use the service to create a new link.

Can we model and exploit these data to steer the online community to a desired activity level? Specifically, can we drive the overall usage of a service to a certain level (*e.g.*, at least twice per day per user) by incentivizing a small number of users to take more initiatives? What if the goal is to make the usage level of a service more homogeneous across users? What about maximizing the overall service usage for a target group of users? Furthermore, these *activity shaping* problems need to be addressed by taking into account budget constraints, since incentives are usually provided in the form of monetary or credit rewards.

Activity shaping problems are significantly more challenging than traditional influence maximization problems, which aim to identify a set of users, who, when convinced to adopt a product, shall influence others in the network and trigger a large cascade of adoptions [1, 2]. First, in influence maximization, the state of each user is often assumed to be binary, either adopting a product or not [1, 3, 4, 5]. However, such assumption does not capture the recurrent nature of product usage, where the frequency of the usage matters. Second, while influence maximization methods identify a set of users to provide incentives, they do not typically provide a quantitative prescription on how much incentive should be provided to each user. Third, activity shaping concerns a larger variety of target states, such as minimum activity and homogeneity of activity, not just activity maximization.

In this paper, we will address the activity shaping problems using multivariate Hawkes processes [6], which can model both endogenous and exogenous recurrent social events, and were shown to be a good fit for such data in a number of recent works (*e.g.*, [7, 8, 9, 10, 11, 12]). More importantly,

we will go beyond model fitting, and derive a novel predictive formula for the overall network activity given the intensity of exogenous events in individual users, using a connection between the processes and branching processes [13, 14, 15, 16]. Based on this relation, we propose a convex optimization framework to address a diverse range of activity shaping problems given budget constraints. Compared to previous methods for influence maximization, our framework can provide more fine-grained control of network activity, not only steering the network to a desired steady-state activity level but also do so in a time-sensitive fashion. For example, our framework allows us to answer complex time-sensitive queries, such as, which users should be incentivized, and by how much, to steer a set of users to use a product twice per week after one month?

In addition to the novel framework, we also develop an efficient gradient based optimization algorithm, where the matrix exponential needed for gradient computation is approximated using the truncated Taylor series expansion [17]. This algorithm allows us to validate our framework in a variety of activity shaping tasks and scale up to networks with tens of thousands of nodes. We also conducted experiments on a network of 60,000 Twitter users and more than 7,500,000 uses of a popular url shortening services. Using held-out data, we show that our algorithm can shape the network behavior much more accurately than alternatives.

## 2    Modeling Endogenous-Exogenous Recurrent Social Events

We model the events generated by $m$ users in a social network as a $m$-dimensional counting process $\boldsymbol{N}(t) = (N_1(t), N_2(t), \ldots, N_m(t))^\top$, where $N_i(t)$ records the total number of events generated by user $i$ up to time $t$. Furthermore, we represent each event as a tuple $(u_i, t_i)$, where $u_i$ is the user identity and $t_i$ is the event timing. Let the history of the process up to time $t$ be $\mathcal{H}_t := \{(u_i, t_i) \,|\, t_i \leqslant t\}$, and $\mathcal{H}_{t-}$ be the history until just before time $t$. Then the increment of the process, $d\boldsymbol{N}(t)$, in an infinitesimal window $[t, t+dt]$ is parametrized by the intensity $\boldsymbol{\lambda}(t) = (\lambda_1(t), \ldots, \lambda_m(t))^\top \geqslant 0$, i.e.,

$$\mathbb{E}[d\boldsymbol{N}(t)|\mathcal{H}_{t-}] = \boldsymbol{\lambda}(t)\,dt. \tag{1}$$

Intuitively, the larger the intensity $\boldsymbol{\lambda}(t)$, the greater the likelihood of observing an event in the time window $[t, t + dt]$. For instance, a Poisson process in $[0, \infty)$ can be viewed as a special counting process with a constant intensity function $\boldsymbol{\lambda}$, independent of time and history. To model the presence of both endogenous and exogenous events, we will decompose the intensity into two terms

$$\underbrace{\boldsymbol{\lambda}(t)}_{\text{overall event intensity}} = \underbrace{\boldsymbol{\lambda}^{(0)}(t)}_{\text{exogenous event intensity}} + \underbrace{\boldsymbol{\lambda}^*(t)}_{\text{endogenous event intensity}}, \tag{2}$$

where the exogenous event intensity models drive outside the network, and the endogenous event intensity models interactions within the network. We assume that hosts of social platforms can potentially drive up or down the exogenous events intensity by providing incentives to users; while endogenous events are generated due to users' own interests or under the influence of network peers, and the hosts do not interfere with them directly. The key questions in the activity shaping context are how to model the endogenous event intensity which are realistic to recurrent social interactions, and how to link the exogenous event intensity to the endogenous event intensity. We assume that the exogenous event intensity is independent of the history and time, i.e., $\boldsymbol{\lambda}^{(0)}(t) = \boldsymbol{\lambda}^{(0)}$.

### 2.1    Multivariate Hawkes Process

Recurrent endogenous events often exhibit the characteristics of self-excitation, where a user tends to repeat what he has been doing recently, and mutual-excitation, where a user simply follows what his neighbors are doing due to peer pressure. These social phenomena have been made analogy to the occurrence of earthquake [18] and the spread of epidemics [19], and can be well-captured by multivariate Hawkes processes [6] as shown in a number of recent works (e.g., [7, 8, 9, 10, 11, 12]).

More specifically, a multivariate Hawkes process is a counting process who has a particular form of intensity. We assume that the strength of influence between users is parameterized by a sparse nonnegative *influence matrix* $\boldsymbol{A} = (a_{uu'})_{u,u' \in [m]}$, where $a_{uu'} > 0$ means user $u'$ directly excites user $u$. We also allow $\boldsymbol{A}$ to have nonnegative diagonals to model self-excitation of a user. Then, the intensity of the $u$-th dimension is

$$\lambda_u^*(t) = \sum_{i:t_i < t} a_{uu_i}\, g(t - t_i) = \sum_{u' \in [m]} a_{uu'} \int_0^t g(t - s)\, dN_{u'}(s), \tag{3}$$

where $g(s)$ is a nonnegative kernel function such that $g(s) = 0$ for $s \leq 0$ and $\int_0^\infty g(s)\,ds < \infty$; the second equality is obtained by grouping events according to users and use the fact that

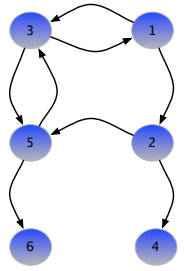
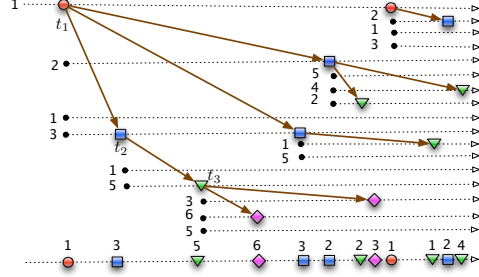

(a) An example social network        (b) Branching structure of events

Figure 1: In Panel (a), each directed edge indicates that the target node *follows*, and can be influenced by, the source node. The activity in this network is modeled using Hawkes processes, which result in branching structure of events shown in Panel (b). Each exogenous event is the root node of a branch (*e.g.*, top left most red circle at $t_1$), and it occurs due to a user's own initiative; and each event can trigger one or more endogenous events (blue square at $t_2$). The new endogenous events can create the next generation of endogenous events (green triangles at $t_3$), and so forth. The social network will constrain the branching structure of events, since an event produced by a user (*e.g.*, user 1) can only trigger endogenous events in the same user or one or more of her followers (*e.g.*, user 2 or 3).

$\int_0^t g(t-s)\,dN_{u'}(s) = \sum_{u_i=u',t_i<t} g(t-t_i)$. Intuitively, $\lambda_u^*(t)$ models the propagation of peer influence over the network — each event $(u_i, t_i)$ occurred in the neighbor of a user will boost her intensity by a certain amount which itself decays over time. Thus, the more frequent the events occur in the user's neighbor, the more likely she will be persuaded to generate a new event.

For simplicity, we will focus on an exponential kernel, $g(t-t_i) = \exp(-\omega(t-t_i))$ in the reminder of the paper. However, multivariate Hawkes processes and the branching processed explained in next section is independent of the kernel choice and can be extended to other kernels such as power-law, Rayleigh or any other long tailed distribution over nonnegative real domain. Furthermore, we can rewrite equation (3) in vectorial format

$$\boldsymbol{\lambda}^*(t) = \int_0^t \boldsymbol{G}(t-s)\,d\boldsymbol{N}(s), \tag{4}$$

by defining a $m \times m$ time-varying matrix $\boldsymbol{G}(t) = (a_{uu'}g(t))_{u,u'\in[m]}$. Note that, for multivariate Hawkes processes, the intensity, $\boldsymbol{\lambda}(t)$, itself is a random quantity, which depends on the history $\mathcal{H}_t$. We denote the expectation of the intensity with respect to history as

$$\boldsymbol{\mu}(t) := \mathbb{E}_{\mathcal{H}_{t-}} [\boldsymbol{\lambda}(t)] \tag{5}$$

### 2.2   Connection to Branching Processes

A branching process is a Markov process that models a population in which each individual in generation $k$ produces some random number of individuals in generation $k+1$, according some distribution [20]. In this section, we will conceptually assign both exogenous events and endogenous events in the multivariate Hawkes process to levels (or generations), and associate these events with a branching structure which records the information on which event triggers which other events (see Figure 1 for an example). Note that this genealogy of events should be interpreted in probabilistic terms and may not be observed in actual data. Such connection has been discussed in Hawkes' original paper on one dimensional Hawkes processes [21], and it has recently been revisited in the context of multivariate Hawkes processes by [11]. The branching structure will play a crucial role in deriving a novel link between the intensity of the exogenous events and the overall network activity.

More specifically, we assign all exogenous events to the zero-th generation, and record the number of such events as $\boldsymbol{N}^{(0)}(t)$. These exogenous events will trigger the first generation of endogenous events whose number will be recorded as $\boldsymbol{N}^{(1)}(t)$. Next these first generation of endogenous events will further trigger a second generation of endogenous events $\boldsymbol{N}^{(2)}(t)$, and so on. Then the total number of events in the network is the sum of the numbers of events from all generations

$$\boldsymbol{N}(t) = \boldsymbol{N}^{(0)}(t) + \boldsymbol{N}^{(1)}(t) + \boldsymbol{N}^{(2)}(t) + \ldots \tag{6}$$

Furthermore, denote all events in generation $k-1$ as $\mathcal{H}_t^{(k-1)}$. Then, independently for each event $(u_i, t_i) \in \mathcal{H}_t^{(k-1)}$ in generation $k-1$, it triggers a Poisson process in its neighbor $u$ independently with intensity $a_{uu_i}g(t-t_i)$. Due to the superposition theorem of independent Poisson processes [22],

the intensity, $\lambda_u^{(k)}(t)$, of events at node $u$ and generation $k$ is simply the sum of conditional intensities of the Poisson processes triggered by all its neighbors, *i.e.*, $\lambda_u^{(k)}(t) = \sum_{(u_i,t_i)\in\mathcal{H}_t^{(k-1)}} a_{uu_i} g(t - t_i) = \sum_{u'\in[m]} \int_0^t g(t-s)\,d\mathbf{N}_{u'}^{(k-1)}(s)$. Concatenate the intensity for all $u \in [m]$, and use the time-varying matrix $\mathbf{G}(t)$ (4), we have

$$\boldsymbol{\lambda}^{(k)}(t) = \int_0^t \mathbf{G}(t-s)\,d\mathbf{N}^{(k-1)}(s), \tag{7}$$

where $\boldsymbol{\lambda}^{(k)}(t) = (\lambda_1^{(k)}(t),\ldots,\lambda_m^{(k)}(t))^\top$ is the intensity for counting process $\mathbf{N}^{(k)}(t)$ at $k$-th generation. Again, due to the superposition of independent Poisson processes, we can decompose the intensity of $\mathbf{N}(t)$ into a sum of conditional intensities from different generation

$$\boldsymbol{\lambda}(t) = \boldsymbol{\lambda}^{(0)}(t) + \boldsymbol{\lambda}^{(1)}(t) + \boldsymbol{\lambda}^{(2)}(t) + \ldots \tag{8}$$

Next, based on the above decomposition, we will develop a closed form relation between the expected intensity $\boldsymbol{\mu}(t) = \mathbb{E}_{\mathcal{H}_{t_-}}[\boldsymbol{\lambda}(t)]$ and the intensity, $\boldsymbol{\lambda}^{(0)}(t)$, of the exogenous events. This relation will form the basis of our activity shaping framework.

## 3 Linking Exogenous Event Intensity to Overall Network Activity

Our strategy is to first link the expected intensity $\boldsymbol{\mu}^{(k)}(t) := \mathbb{E}_{\mathcal{H}_{t_-}}[\boldsymbol{\lambda}^{(k)}(t)]$ of events at the $k$-th generation with $\boldsymbol{\lambda}^{(0)}(t)$, and then derive a close form for the infinite series sum

$$\boldsymbol{\mu}(t) = \boldsymbol{\mu}^{(0)}(t) + \boldsymbol{\mu}^{(1)}(t) + \boldsymbol{\mu}^{(2)}(t) + \ldots \tag{9}$$

Define a series of auto-convolution matrices, one for each generation, with $\mathbf{G}^{(\star 0)}(t) = \mathbf{I}$ and

$$\mathbf{G}^{(\star k)}(t) = \int_0^t \mathbf{G}(t-s)\,\mathbf{G}^{(\star k-1)}(s)\,ds = \mathbf{G}(t) \star \mathbf{G}^{(\star k-1)}(t) \tag{10}$$

Then the expected intensity of events at the $k$-th generation is related to exogenous intensity $\boldsymbol{\lambda}^{(0)}$ by

**Lemma 1** $\boldsymbol{\mu}^{(k)}(t) = \mathbf{G}^{(\star k)}(t)\,\boldsymbol{\lambda}^{(0)}$.

Next, by summing together all auto-convolution matrices,

$$\boldsymbol{\Psi}(t) := \mathbf{I} + \mathbf{G}^{(\star 1)}(t) + \mathbf{G}^{(\star 2)}(t) + \ldots$$

we obtain a linear relation between the expected intensity of the network and the intensity of the exogenous events, *i.e.*, $\boldsymbol{\mu}(t) = \boldsymbol{\Psi}(t)\boldsymbol{\lambda}^{(0)}$. The entries in the matrix $\boldsymbol{\Psi}(t)$ roughly encode the "influence" between pairs of users. More precisely, the entry $\boldsymbol{\Psi}_{uv}(t)$ is the expected intensity of events at node $u$ due to a unit level of exogenous intensity at node $v$. We can also derive several other useful quantities from $\boldsymbol{\Psi}(t)$. For example, $\boldsymbol{\Psi}_{\bullet v}(t) := \sum_u \boldsymbol{\Psi}_{uv}(t)$ can be thought of as the overall influence user $v$ has on all users. Surprisingly, for exponential kernel, the infinite sum of matrices results in a closed form using matrix exponentials. First, let $\widehat{\cdot}$ denote the Laplace transform of a function, and we have the following intermediate results on the Laplace transform of $\mathbf{G}^{(\star k)}(t)$.

**Lemma 2** $\widehat{\mathbf{G}}^{(\star k)}(z) = \int_0^\infty \mathbf{G}^{(\star k)}(t)\,dt = \frac{1}{z} \cdot \frac{\mathbf{A}^k}{(z+\omega)^k}$

With Lemma 2, we are in a position to prove our main theorem below:

**Theorem 3** $\boldsymbol{\mu}(t) = \boldsymbol{\Psi}(t)\boldsymbol{\lambda}^{(0)} = \left(e^{(\mathbf{A}-\omega\mathbf{I})t} + \omega(\mathbf{A}-\omega\mathbf{I})^{-1}(e^{(\mathbf{A}-\omega\mathbf{I})t} - \mathbf{I})\right)\boldsymbol{\lambda}^{(0)}$.

Theorem 3 provides us a linear relation between exogenous event intensity and the expected overall intensity at any point in time but not just stationary intensity. The significance of this result is that it allows us later to design a diverse range of convex programs to determine the intensity of the exogenous event in order to achieve a target intensity.

In fact, we can recover the previous results in the stationary case as a special case of our general result. More specifically, a multivariate Hawkes process is stationary if the spectral radius

$$\boldsymbol{\Gamma} := \int_0^\infty \mathbf{G}(t)\,dt = \left(\int_0^\infty g(t)\,dt\right)\left(a_{uu'}\right)_{u,u'\in[m]} = \frac{\mathbf{A}}{\omega} \tag{11}$$

is strictly smaller than 1 [6]. In this case, the expected intensity is $\boldsymbol{\mu} = (\mathbf{I} - \boldsymbol{\Gamma})^{-1}\boldsymbol{\lambda}^{(0)}$ independent of the time. We can obtain this relation from theorem 3 if we let $t \to \infty$.

**Corollary 4** $\boldsymbol{\mu} = (\mathbf{I} - \boldsymbol{\Gamma})^{-1}\boldsymbol{\lambda}^{(0)} = \lim_{t\to\infty} \boldsymbol{\Psi}(t)\,\boldsymbol{\lambda}^{(0)}$.

Refer to Appendix A for all the proofs.

## 4 Convex Activity Shaping Framework

Given the linear relation between exogenous event intensity and expected overall event intensity, we now propose a convex optimization framework for a variety of activity shaping tasks. In all tasks discussed below, we will optimize the exogenous event intensity $\boldsymbol{\lambda}^{(0)}$ such that the expected overall event intensity $\boldsymbol{\mu}(t)$ is maximized with respect to some concave utility $U(\cdot)$ in $\boldsymbol{\mu}(t)$, *i.e.*,

$$\begin{aligned}\text{maximize}_{\boldsymbol{\mu}(t),\boldsymbol{\lambda}^{(0)}} \quad & U(\boldsymbol{\mu}(t)) \\ \text{subject to} \quad & \boldsymbol{\mu}(t) = \boldsymbol{\Psi}(t)\boldsymbol{\lambda}^{(0)}, \quad \boldsymbol{c}^\top \boldsymbol{\lambda}^{(0)} \leqslant C, \quad \boldsymbol{\lambda}^{(0)} \geqslant 0\end{aligned} \tag{12}$$

where $\boldsymbol{c} = (c_1, \ldots, c_m)^\top \geqslant 0$ is the cost per unit event for each user and $C$ is the total budget. Additional regularization can also be added to $\boldsymbol{\lambda}^{(0)}$ either to restrict the number of incentivized users (with $\ell_0$ norm $\|\boldsymbol{\lambda}^{(0)}\|_0$), or to promote a sparse solution (with $\ell_1$ norm $\|\boldsymbol{\lambda}^{(0)}\|_1$, or to obtain a smooth solution (with $\ell_2$ regularization $\|\boldsymbol{\lambda}^{(0)}\|_2$). We next discuss several instances of the general framework which achieve different goals (their constraints remain the same and hence omitted).

**Capped Activity Maximization.** In real networks, there is an upper bound (or a cap) on the activity each user can generate due to limited attention of a user. For example, a Twitter user typically posts a limited number of shortened urls or retweets a limited number of tweets [23]. Suppose we know the upper bound, $\alpha_u$, on a user's activity, *i.e.*, how much activity each user is willing to generate. Then we can perform the following *capped activity maximization* task

$$\text{maximize}_{\boldsymbol{\mu}(t),\boldsymbol{\lambda}^{(0)}} \quad \sum_{u\in[m]} \min\{\mu_u(t), \alpha_u\} \tag{13}$$

**Minimax Activity Shaping.** Suppose our goal is instead maintaining the activity of each user in the network above a certain minimum level, or, alternatively make the user with the minimum activity as active as possible. Then, we can perform the following *minimax activity shaping* task

$$\text{maximize}_{\boldsymbol{\mu}(t),\boldsymbol{\lambda}^{(0)}} \quad \min_u \ \mu_u(t) \tag{14}$$

**Least-Squares Activity Shaping.** Sometimes we want to achieve a pre-specified target activity levels, $\boldsymbol{v}$, for users. For example, we may like to divide users into groups and desire a different level of activity in each group. Inspired by these examples, we can perform the following *least-squares activity shaping* task

$$\text{maximize}_{\boldsymbol{\mu}(t),\boldsymbol{\lambda}^{(0)}} \quad -\|\boldsymbol{B}\boldsymbol{\mu}(t) - \boldsymbol{v}\|_2^2 \tag{15}$$

where $\boldsymbol{B}$ encodes potentially additional constraints (*e.g.*, group partitions). Besides Euclidean distance, the family of Bregman divergences can be used to measure the difference between $\boldsymbol{B}\boldsymbol{\mu}(t)$ and $\boldsymbol{v}$ here. That is, given a function $f(\cdot) : \mathbb{R}^m \mapsto \mathbb{R}$ convex in its argument, we can use $D(\boldsymbol{B}\boldsymbol{\mu}(t)\|\boldsymbol{v}) := f(\boldsymbol{B}\boldsymbol{\mu}(t)) - f(\boldsymbol{v}) - \langle \nabla f(\boldsymbol{v}), \boldsymbol{B}\boldsymbol{\mu}(t) - \boldsymbol{v}\rangle$ as our objective function.

**Activity Homogenization.** Many other concave utility functions can be used. For example, we may want to steer users activities to a more homogeneous profile. If we measure homogeneity of activity with Shannon entropy, then we can perform the following activity homogenization task

$$\text{maximize}_{\boldsymbol{\mu}(t),\boldsymbol{\lambda}^{(0)}} \quad -\sum_{u\in[m]} \mu_u(t)\ln\mu_u(t) \tag{16}$$

## 5 Scalable Algorithm

All the activity shaping problems defined above require an efficient evaluation of the instantaneous average intensity $\boldsymbol{\mu}(t)$ at time $t$, which entails computing matrix exponentials to obtain $\boldsymbol{\Psi}(t)$. In small or medium networks, we can rely on well-known numerical methods to compute matrix exponentials [24]. However, in large networks, the explicit computation of $\boldsymbol{\Psi}(t)$ becomes intractable.

Fortunately, we can exploit the following key property of our convex activity shaping framework: the instantaneous average intensity only depends on $\boldsymbol{\Psi}(t)$ through matrix-vector product operations. In particular, we start by using Theorem 3 to rewrite the multiplication of $\boldsymbol{\Psi}(t)$ and a vector $\boldsymbol{v}$ as $\boldsymbol{\Psi}(t)\boldsymbol{v} = e^{(\boldsymbol{A}-\omega\boldsymbol{I})t}\boldsymbol{v} + \omega(\boldsymbol{A} - \omega\boldsymbol{I})^{-1}\left(e^{(\boldsymbol{A}-\omega\boldsymbol{I})t}\boldsymbol{v} - \boldsymbol{v}\right)$. We then get a tractable solution by first computing $e^{(\boldsymbol{A}-\omega\boldsymbol{I})t}\boldsymbol{v}$ efficiently, subtracting $\boldsymbol{v}$ from it, and solving a sparse linear system of equations, $(\boldsymbol{A} - \omega\boldsymbol{I})x = \left(e^{(\boldsymbol{A}-\omega\boldsymbol{I})t}\boldsymbol{v} - \boldsymbol{v}\right)$, efficiently. The steps are illustrated in Algorithm 1. Next, we elaborate on two very efficient algorithms for computing the product of matrix exponential with a vector and for solving a sparse linear system of equations.

For the computation of the product of matrix exponential with a vector, we rely on the iterative algorithm by Al-Mohy et al. [17], which combines a scaling and squaring method with a truncated Taylor series approximation to the matrix exponential. For solving the sparse linear system of equa-

| **Algorithm 1:** Average Instantaneous Intensity | **Algorithm 2:** PGD for Activity Shaping |
|---|---|
| **input** : $\boldsymbol{A}, \omega, t, \boldsymbol{v}$ <br> **output**: $\boldsymbol{\Psi}(t)\boldsymbol{v}$ <br> $\boldsymbol{v}_1 = e^{(\boldsymbol{A}-\omega\boldsymbol{I})t}\boldsymbol{v}$ <br> $\boldsymbol{v}_2 = \boldsymbol{v}_2 - \boldsymbol{v}$; <br> $\boldsymbol{v}_3 = (\boldsymbol{A}-\omega\boldsymbol{I})^{-1}\boldsymbol{v}_2$ <br> **return** $\boldsymbol{v}_1 + \omega\boldsymbol{v}_3$; | Initialize $\boldsymbol{\lambda}^{(0)}$; <br> **repeat** <br>     1- Project $\boldsymbol{\lambda}^{(0)}$ into $\boldsymbol{\lambda}^{(0)} \geqslant 0$, $\boldsymbol{c}^\top\boldsymbol{\lambda}^{(0)} \leqslant C$; <br>     2- Evaluate the gradient $\boldsymbol{g}(\boldsymbol{\lambda}^{(0)})$ at $\boldsymbol{\lambda}^{(0)}$; <br>     3- Update $\boldsymbol{\lambda}^{(0)}$ using the gradient $\boldsymbol{g}(\boldsymbol{\lambda}^{(0)})$; <br> **until** *convergence*; |

tion, we use the well-known GMRES method [25], which is an Arnoldi process for constructing an $l_2$-orthogonal basis of Krylov subspaces. The method solves the linear system by iteratively minimizing the norm of the residual vector over a Krylov subspace.

Perhaps surprisingly, we will now show that it is possible to compute the gradient of the objective functions of all our activity shaping problems using the algorithm developed above for computing the average instantaneous intensity. We only need to define the vector $\boldsymbol{v}$ appropriately for each problem, as follows: *(i)* Activity maximization: $\boldsymbol{g}(\boldsymbol{\lambda}^{(0)}) = \boldsymbol{\Psi}(t)^\top\boldsymbol{v}$, where $\boldsymbol{v}$ is defined such that $v_j = 1$ if $\alpha_j > \mu_j$, and $v_j = 0$, otherwise. *(ii)* Minimax activity shaping: $\boldsymbol{g}(\boldsymbol{\lambda}^{(0)}) = \boldsymbol{\Psi}(t)^\top\boldsymbol{e}$, where $\boldsymbol{e}$ is defined such that $e_j = 1$ if $\mu_j = \mu_{min}$, and $e_j = 0$, otherwise. *(iii)* Least-squares activity shaping: $\boldsymbol{g}(\boldsymbol{\lambda}^{(0)}) = 2\boldsymbol{\Psi}(t)^\top\boldsymbol{B}^\top\left(\boldsymbol{B}\boldsymbol{\Psi}(t)\boldsymbol{\lambda}^{(0)} - \boldsymbol{v}\right)$. *(iv)* Activity homogenization: $\boldsymbol{g}(\boldsymbol{\lambda}^{(0)}) = \boldsymbol{\Psi}(t)^\top\ln\left(\boldsymbol{\Psi}(t)\boldsymbol{\lambda}^{(0)}\right) + \boldsymbol{\Psi}(t)^\top\boldsymbol{1}$, where $\ln(\cdot)$ on a vector is the element-wise natural logarithm. Since the activity maximization and the minimax activity shaping tasks require only one evaluation of $\boldsymbol{\Psi}(t)$ times a vector, Algorithm 1 can be used directly. However, computing the gradient for least-squares activity shaping and activity homogenization is slightly more involved and it requires to be careful with the order in which we perform the operations (Refer to Appendix B for details). Equipped with an efficient way to compute of gradients, we solve the corresponding convex optimization problem for each activity shaping problem by applying projected gradient descent (PGD) [26] with the appropriate gradient[1]. Algorithm 2 summarizes the key steps.

## 6   Experimental Evaluation

We evaluate our framework using both simulated and real world held-out data, and show that our approach significantly outperforms several baselines. The appendix contains additional experiments.

**Dataset description and network inference.** We use data gathered from Twitter as reported in [27], which comprises of all public tweets posted by 60,000 users during a 8-month period, from January 2009 to September 2009. For every user, we record the times she uses any of six popular url shortening services (refer to Appendix C for details). We evaluate the performance of our framework on a subset of 2,241 active users, linked by 4,901 edges, which we call 2K dataset, and we evaluate its scalability on the overall 60,000 users, linked by $\sim 200,000$ edges, which we call 60K dataset. The 2K dataset accounts for 691,020 url shortened service uses while the 60K dataset accounts for $\sim 7.5$ million uses. Finally, we treat each service as independent cascades of events.

In the experiments, we estimated the nonnegative influence matrix $\boldsymbol{A}$ and the exogenous intensity $\boldsymbol{\lambda}^{(0)}$ using maximum log-likelihood, as in previous work [8, 9, 12]. We used a temporal resolution of one minute and selected the bandwidth $\omega = 0.1$ by cross validation. Loosely speaking, $\omega = 0.1$ corresponds to loosing 70% of the initial influence after 10 minutes, which may be explained by the rapid rate at which each user' news feed gets updated.

**Evaluation schemes.** We focus on three tasks: capped activity maximization, minimax activity shaping, and least square activity shaping. We set the total budget to $C = 0.5$, which corresponds to supporting a total extra activity equal to $0.5$ actions per unit time, and assume all users entail the same cost. In the capped activity maximization, we set the upper limit of each user's intensity, $\boldsymbol{\alpha}$, by adding a nonnegative random vector to their inferred initial intensity. In the least-squares activity shaping, we set $\boldsymbol{B} = \boldsymbol{I}$ and aim to create three user groups: less-active, moderate, and super-active. We use three different evaluation schemes, with an increasing resemblance to a real world scenario:

*Theoretical objective*: We compute the expected overall (theoretical) intensity by applying Theorem 3 on the optimal exogenous event intensities to each of the three activity shaping tasks, as well as the learned $\boldsymbol{A}$ and $\omega$. We then compute and report the value of the objective functions.

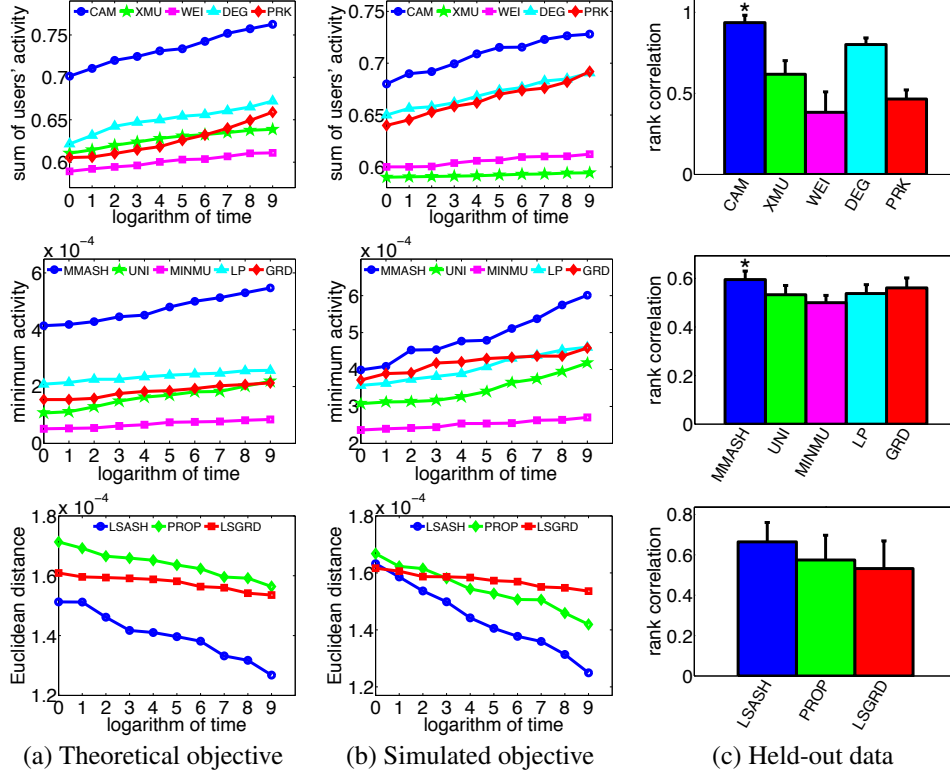

(a) Theoretical objective      (b) Simulated objective      (c) Held-out data

Figure 2: Row 1: Capped activity maximization. Row 2: Minimax activity shaping. Row 3: Least-squares activity shaping. * means statistical significant at level of 0.01 with paired t-test between our method and the second best

*Simulated objective*: We simulate 50 cascades with Ogata's thinning algorithm [28], using the optimal exogenous event intensities to each of the three activity shaping tasks, and the learned $A$ and $\omega$. We then estimate empirically the overall event intensity based on the simulated cascades, by computing a running average over non-overlapping time windows, and report the value of the objective functions based on this estimated overall intensity. Appendix D provides a comparison between the simulated and the theoretical objective.

*Held-out data*: The most interesting evaluation scheme would entail carrying out real interventions in a social platform. However, since this is very challenging to do, instead, in this evaluation scheme, we use held-out data to simulate such process, proceeding as follows. We first partition the 8-month data into 50 five-day long contiguous intervals. Then, we use one interval for training and the remaining 49 intervals for testing. Suppose interval 1 is used for training, the procedure is as follows:

1. We estimate $A_1$, $\omega_1$ and $\lambda_1^{(0)}$ using the events from interval 1. Then, we fix $A_1$ and $\omega_1$, and estimate $\lambda_i^{(0)}$ for all other intervals, $i = 2, \ldots, 49$.
2. Given $A_1$ and $\omega_1$, we find the optimal exogenous event intensities, $\lambda_{opt}^{(0)}$, for each of the three activity shaping task, by solving the associated convex program. We then sort the estimated $\lambda_i^{(0)}$ ($i = 2, \ldots, 49$) according to their similarity to $\lambda_{opt}^{(0)}$, using the Euclidean distance $\|\lambda_{opt}^{(0)} - \lambda_i^{(0)}\|_2$.
3. We estimate the overall event intensity for each of the 49 intervals ($i = 2, \ldots, 49$), as in the "simulated objective" evaluation scheme, and sort these intervals according to the value of their corresponding objective function.
4. Last, we compute and report the rank correlation score between the two orderings obtained in step 2 and 3.[2] The larger the rank correlation, the better the method.

We repeat this procedure 50 times, choosing each different interval for training once, and compute and report the average rank correlations. More details can be found in the appendix.

**Capped activity maximization (CAM).** We compare to a number of alternatives. XMU: heuristic based on $\boldsymbol{\mu}(t)$ without optimization; DEG and WEI: heuristics based on the degree of the user; PRANK: heuristic based on page rank (refer to Appendix C for further details). The first row of Figure 2 summarizes the results for the three different evaluation schemes. We find that our method (CAM) consistently outperforms the alternatives. For the theoretical objective, CAM is 11 % better than the second best, DEG. The difference in overall users' intensity from DEG is about $0.8$ which, roughly speaking, leads to at least an increase of about $0.8 \times 60 \times 24 \times 30 = 34,560$ in the overall number of events in a month. In terms of simulated objective and held-out data, the results are similar and provide empirical evidence that, compared to other heuristics, degree is an appropriate surrogate for influence, while, based on the poor performance of XMU, it seems that high activity does not necessarily entail being influential. To elaborate on the interpretability of the real-world experiment on held-out data, consider for example the difference in rank correlation between CAM and DEG, which is almost $0.1$. Then, roughly speaking, this means that incentivizing users based on our approach accommodates with the ordering of real activity patterns in $0.1 \times \frac{50 \times 49}{2} = 122.5$ more pairs of realizations.

**Minimax activity shaping (MMASH).** We compare to a number of alternatives. UNI: heuristic based on equal allocation; MINMU: heuristic based on $\boldsymbol{\mu}(t)$ without optimization; LP: linear programming based heuristic; GRD: a greedy approach to leverage the activity (see Appendix C for more details). The second row of Figure 2 summarizes the results for the three different evaluation schemes. We find that our method (MMASH) consistently outperforms the alternatives. For the theoretical objective, it is about $2\times$ better than the second best, LP. Importantly, the difference between MMASH and LP is not trifling and the least active user carries out $2 \times 10^{-4} \times 60 \times 24 \times 30 = 4.3$ more actions in average over a month. As one may have expected, GRD and LP are the best among the heuristics. The poor performance of MINMU, which is directly related to the objective of MMASH, may be because it assigns the budget to a low active user, regardless of their influence. However, our method, by cleverly distributing the budget to the users whom actions trigger many other users' actions (like those ones with low activity), it benefits from the budget most. In terms of simulated objective and held-out data, the algorithms' performance become more similar.

**Least-squares activity shaping (LSASH).** We compare to two alternatives. PROP: Assigning the budget proportionally to the desired activity; LSGRD: greedily allocating budget according the difference between current and desired activity (refer to Appendix C for more details). The third row of Figure 2 summarizes the results for the three different evaluation schemes. We find that our method (LSASH) consistently outperforms the alternatives. Perhaps surprisingly, PROP, despite its simplicity, seems to perform slightly better than LSGRD. This is may be due to the way it allocates the budget to users, *e.g.*, it does not aim to strictly fulfill users' target activity but benefit more users by assigning budget proportionally. Refer to Appendix E for additional experiments.

**Sparsity and Activity Shaping.** In some applications there is a limitation on the number of users we can incentivize. In our proposed framework, we can handle this requirement by including a sparsity constraint on the optimization problem. In order to maintain the convexity of the optimization problem, we consider a $l_1$ regularization term, where a regularization parameter $\gamma$ provides the trade-off between sparsity and the activity shaping goal. Refer to Appendix F for more details and experimental results for different values of $\gamma$.

**Scalability.** The most computationally demanding part of the proposed algorithm is the evaluation of matrix exponentials, which we scale up by utilizing techniques from matrix algebra, such as GMRES and Al-Mohy methods. As a result, we are able to run our methods in a reasonable amount of time on the 60K dataset, specifically, in comparison with a naive implementation of matrix exponential evaluations. Refer to Appendix G for detailed experimental results on scalability.

Appendix H discusses the limitations of our framework and future work.

**Acknowledgement.** This project was supported in part by NSF IIS1116886, NSF/NIH BIGDATA 1R01GM108341, NSF CAREER IIS1350983 and Raytheon Faculty Fellowship to Le Song. Isabel Valera acknowledge the support of *Plan Regional-Programas I+D* of *Comunidad de Madrid* (AGES-CM S2010/BMD-2422), *Ministerio de Ciencia e Innovación* of Spain (project DEIPRO TEC2009-14504-C02-00 and program Consolider-Ingenio 2010 CSD2008-00010 COMONSENS).

## Footnotes

[1]For nondifferential objectives, subgradient algorithms can be used instead.

[2] rank correlation = number of pairs with consistent ordering / total number of pairs.

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
