[Supplementary Material]

# A  Proofs

**Lemma 1** $\boldsymbol{\mu}^{(k)}(t) = \boldsymbol{G}^{(\star k)}(t)\,\boldsymbol{\lambda}^{(0)}$.

**Proof** We will prove the lemma by induction. For generation $k = 0$, $\boldsymbol{\mu}^{(0)}(t) = \mathbb{E}_{\mathcal{H}_t}[\boldsymbol{\lambda}^{(0)}] = \boldsymbol{G}^{(\star 0)}(t)\boldsymbol{\lambda}^{(0)}$. Assume the relation holds for generation $k$: $\boldsymbol{\mu}^{(k)}(t) = \boldsymbol{G}^{(\star k)}(t)\boldsymbol{\lambda}^{(0)}$. Then for generation $k+1$, we have $\boldsymbol{\mu}^{(k+1)}(t) = \mathbb{E}_{\mathcal{H}_t}[\int_0^t \boldsymbol{G}(t-s)\,d\boldsymbol{N}^{(k)}(s)] = \int_0^t \boldsymbol{G}(t-s)\,\mathbb{E}_{\mathcal{H}_t}[d\boldsymbol{N}^{(k)}(s)]$. By definition $\mathbb{E}_{\mathcal{H}_t}[d\boldsymbol{N}^{(k)}(s)] = \mathbb{E}_{\mathcal{H}_{s-}}[\mathbb{E}[d\boldsymbol{N}^{(k)}(s)|\mathcal{H}_{s-}]] = \mathbb{E}_{\mathcal{H}_{s-}}[\boldsymbol{\lambda}^{(k)}(s)\,ds] = \boldsymbol{\mu}^{(k)}(s)\,ds$, then substitute it in and we have

$$\boldsymbol{\mu}^{(k+1)}(t) = \int_0^t \boldsymbol{G}(t-s)\,\boldsymbol{G}^{(\star k)}(s)\,\boldsymbol{\lambda}^{(0)}\,ds = \boldsymbol{G}^{(\star k+1)}(t)\boldsymbol{\lambda}^{(0)},$$

which completes the proof. ∎

**Lemma 2** $\widehat{\boldsymbol{G}}^{(\star k)}(z) = \int_0^\infty \boldsymbol{G}^{(\star k)}(t)\,dt = \frac{1}{z} \cdot \frac{\boldsymbol{A}^k}{(z+\omega)^k}$

**Proof** We will prove the result by induction on $k$. First, given our choice of exponential kernel, $\boldsymbol{G}(t) = e^{-\omega t}\boldsymbol{A}$, we have that $\widehat{\boldsymbol{G}}(z) = \frac{1}{z+w}\boldsymbol{A}$. Then for $k = 0$, $\boldsymbol{G}^{(\star 0)}(t) = \boldsymbol{I}$ and $\widehat{\boldsymbol{I}}(z) = \int_0^\infty e^{-zt}\boldsymbol{I}\,dt = \frac{1}{z}\boldsymbol{I}$. Now assume the result hold for a general $k-1$, then $\widehat{\boldsymbol{G}}^{(\star k-1)}(z) = \frac{1}{z} \cdot \frac{\boldsymbol{A}^{k-1}}{(z+\omega)^{k-1}}$. Next, for $k$, we have $\widehat{\boldsymbol{G}}^{(\star k)}(z) = \int_0^\infty e^{-zt}\left(\boldsymbol{G}(t) \star \boldsymbol{G}^{(\star k-1)}(t)\right)dt = \widehat{\boldsymbol{G}}(z)\widehat{\boldsymbol{G}}^{(\star k-1)}(z)$, which is $\left(\frac{1}{z+\omega}\boldsymbol{A}\right)\left(\frac{\boldsymbol{A}^{k-1}}{(z+\omega)^{k-1}} \cdot \frac{1}{z}\right) = \frac{1}{z} \cdot \frac{\boldsymbol{A}^k}{(z+\omega)^k}$, and completes the proof. ∎

**Theorem 3** $\boldsymbol{\mu}(t) = \boldsymbol{\Psi}(t)\boldsymbol{\lambda}^{(0)} = \left(e^{(\boldsymbol{A}-\omega\boldsymbol{I})t} + \omega(\boldsymbol{A} - \omega\boldsymbol{I})^{-1}(e^{(\boldsymbol{A}-\omega\boldsymbol{I})t} - \boldsymbol{I})\right)\boldsymbol{\lambda}^{(0)}$.

**Proof** We first compute the Laplace transform $\widehat{\boldsymbol{\Psi}}(z) := \int_0^\infty e^{-zt}\boldsymbol{\Psi}(t)\,dt$. Using lemma 2, we have

$$\widehat{\boldsymbol{\Psi}}(z) = \frac{1}{z}\sum_{i=0}^\infty \frac{\boldsymbol{A}^i}{(z+w)^i} = \frac{(z+w)}{z}\sum_{i=0}^\infty \frac{\boldsymbol{A}^{i+1}}{(z+w)^i}$$

Second, let $\widehat{\boldsymbol{F}}(z) := \sum_{i=0}^\infty \frac{\boldsymbol{A}^i}{z^{i+1}}$ and its inverse Laplace transform be $\boldsymbol{F}(t) = \int_0^\infty e^{zt}\widehat{\boldsymbol{F}}(z)\,dz = \sum_{i=0}^\infty \frac{(\boldsymbol{A}t)^i}{i!} = e^{\boldsymbol{A}t}$, where $e^{\boldsymbol{A}t}$ is a matrix exponential. Then, it is easy to see that $\widehat{\boldsymbol{\Psi}}(z) = \frac{(z+w)}{z}\widehat{\boldsymbol{F}}(z+w) = \widehat{\boldsymbol{F}}(z+w) + \frac{w}{z}\widehat{\boldsymbol{F}}(z+w)$. Finally, we perform inverse Laplace transform for $\widehat{\boldsymbol{\Psi}}(z)$, and obtain $\boldsymbol{\Psi}(t) = e^{(\boldsymbol{A}-\omega\boldsymbol{I})t} + \omega\int_0^t e^{(\boldsymbol{A}-\omega\boldsymbol{I})s}ds = e^{(\boldsymbol{A}-\omega\boldsymbol{I})t} + \omega(\boldsymbol{A} - \omega\boldsymbol{I})^{-1}\left(e^{(\boldsymbol{A}-\omega\boldsymbol{I})t} - \boldsymbol{I}\right)$, where we made use of the property of Laplace transform that dividing by $z$ in the frequency domain is equal to an integration in time domain, and $F(z+w) = e^{-\omega t}e^{\boldsymbol{A}t} = e^{(\boldsymbol{A}-\omega\boldsymbol{I})t}$. ∎

**Corollary 4** $\boldsymbol{\mu} = (\boldsymbol{I} - \boldsymbol{\Gamma})^{-1}\boldsymbol{\lambda}^{(0)} = \lim_{t\to\infty}\boldsymbol{\Psi}(t)\,\boldsymbol{\lambda}^{(0)}$.

**Proof** If the process is stationary, the spectral radius of $\boldsymbol{\Gamma} = \frac{\boldsymbol{A}}{w}$ is smaller than 1, which implies that all eigenvalues of $\boldsymbol{A}$ are smaller than $\omega$ in magnitude. Thus, all eigenvalues of $\boldsymbol{A} - \omega\boldsymbol{I}$ are negative. Let $\boldsymbol{P}\boldsymbol{D}\boldsymbol{P}^{-1}$ be the eigenvalue decomposition of $\boldsymbol{A} - \omega\boldsymbol{I}$, and all the elements (in diagonal) of $\boldsymbol{D}$ are negative. Then based on the property of matrix exponential, we have $e^{(\boldsymbol{A}-\omega\boldsymbol{I})t} = \boldsymbol{P}e^{\boldsymbol{D}t}\boldsymbol{P}^{-1}$. As we let $t \to \infty$, the matrix $e^{\boldsymbol{D}t} \to \boldsymbol{0}$ and hence $e^{(\boldsymbol{A}-\omega\boldsymbol{I})t} \to \boldsymbol{0}$. Thus $\lim_{t\to\infty}\boldsymbol{\Psi}(t) = -\omega(\boldsymbol{A} - \omega\boldsymbol{I})^{-1}$, which is equal to $(\boldsymbol{I} - \boldsymbol{\Gamma})^{-1}$, and completes the proof. ∎

---

**Algorithm 3:** Gradient For Least-squares Activity Shaping

---

**input** : $\boldsymbol{A}, \omega, t, \boldsymbol{v}, \boldsymbol{\lambda}^{(0)}$
**output**: $\boldsymbol{g}(\boldsymbol{\lambda}^{(0)})$
$\boldsymbol{v}_1 = \boldsymbol{\Psi}(t)\boldsymbol{\lambda}^{(0)}$ ;        //Application of algorithm 1
$\boldsymbol{v}_2 = \boldsymbol{B}\boldsymbol{v}_1$ ;      //Sparse matrix vector product
$\boldsymbol{v}_3 = \boldsymbol{B}^\top(\boldsymbol{v}_2 - \boldsymbol{v})$ ;        //Sparse matrix vector product
$\boldsymbol{v}_4 = \boldsymbol{\Psi}(t)\boldsymbol{v}_3$ ;      //Application of algorithm 1
**return** $2\boldsymbol{v}_4$

---

---

**Algorithm 4:** Gradient For Activity Homogenization

---

**input** : $\boldsymbol{A}, \omega, t, \boldsymbol{v}, \boldsymbol{\lambda}^{(0)}$
**output**: $\boldsymbol{g}(\boldsymbol{\lambda}^{(0)})$
$\boldsymbol{v}_1 = \boldsymbol{\Psi}(t)\boldsymbol{\lambda}^{(0)}$ ;        //Application of algorithm 1
$\boldsymbol{v}_2 = \ln(\boldsymbol{v}_1)$;
$\boldsymbol{v}_3 = \boldsymbol{\Psi}(t)^\top \boldsymbol{v}_2$ ;        //Application of algorithm 1
$\boldsymbol{v}_4 = \boldsymbol{\Psi}(t)^\top \boldsymbol{1}$ ;        //Application of algorithm 1
**return** $\boldsymbol{v}_3 + \boldsymbol{v}_4$

---

## B    Gradients for Least-Square Activity Shaping and Activity Homogenization

Algorithm 3 includes the efficient procedure to compute the gradient in the least-squares activity shaping task. Since $\boldsymbol{B}$ is usually sparse, it includes two multiplications of a sparse matrix and a vector, two matrix exponentials multiplied by a vector, and two sparse linear systems of equations. Algorithm 4 summarizes the steps for efficient computation of the gradient in the activity homogenization task. Assuming again a sparse $\boldsymbol{B}$, it consists of two multiplication of a matrix exponential and a vector and two sparse linear systems of equations.

## C    More on the Experimental Setup

Table 1 shows the number of adopters and usages for the six different URL shortening services. It includes a total of 7,566,098 events (adoptions) during the 8-month period.

Next, we describe the considered baselines proposed to compare to our approach for i) the capped activity maximization; ii) the minimax activity shaping; and iii) the least-squares activity shaping problems.

For *capped activity maximization* problem, we consider the following four heuristic baselines:

- XMU allocates the budget based on users' current activity. In particular, it assigns the budget to each of the half top-most active users proportional to their average activity, $\boldsymbol{\mu}(t)$, computed from the inferred parameters.

- WEI assigns positive budget to the users proportionally to their sum of out-going influence ($\sum_u a_{uu'}$). This heuristic allows us (by comparing its results to CAM) to understand the effect of considering the whole network with respect to only consider the direct (out-going) influence.

- DEG assumes that more central users, *i.e.*, more connected users, can leverage the total activity, therefore, assigns the budget to the more connected users proportional to their degree in the network.

- PRK sorts the users according to the their pagerank in the weighted influence network ($\boldsymbol{A}$) with the damping factor set to $0.85\%$, and assigns the budget to the top users proportional their pagerank value.

In order to show how network structure leverages the *minimax activity shaping* we implement following four baselines:

| Service | # adopters | # usages |
|---------|-----------|----------|
| Bitly | 55,883 | 5,046,710 |
| TinyURL | 46,577 | 1,682,459 |
| Isgd | 28,050 | 596,895 |
| TwURL | 15,215 | 197,568 |
| SnURL | 4,462 | 41,823 |
| Doiop | 88 | 643 |

Table 1: # of adopters and usages for each URL shortening service.

- UNI allocates the total budget equally to all users.
- MINMU divides uniformly the total budget among half of the users with lower average activity $\boldsymbol{\mu}(t)$, which is computed from the inferred parameters.
- LP finds the top half of least-active users in the current network and allocates the budget such that after the assignment the network has the highest minimum activity possible. This method uses linear programming to learn exogenous activity of the users, but, in contrast to the proposed method, does not consider the network and propagation of adoptions.
- GRD finds the user with minimum activity, assigns a portion of the budget, and computes the resulting $\boldsymbol{\mu}(t)$. It then repeats the process to incentivize half of users.

We compare ***least-square activity shaping*** with the following baselines:

- PROP shapes the activity by allocating the budget proportional to the desired shape, *i.e.*, the shape of the assignment is similar to the target shape.
- LSGRD greedily finds the user with the highest distance between her current and target activity, assigns her a budget to reach her target, and proceeds this way to consume the whole budget.

Each baseline relies on a specific property to allocate the budget (*e.g.* connectedness in DEG). However, most of them face two problems: The first one is how many users to incentivize and the second one is how much should be paid to the selected users. They usually rely on heuristics to reveal these two problems (*e.g.* allocating an amount proportional to that property and/or to the top half users sorted based on the specific property). In contrast, our framework is comprehensive enough to address those difficulties based on well-developed theoretical basis. This key factor accompanied with the appropriate properties of Hawkes process for modeling social influence (*e.g.* mutually exciting) make the proposed method the best.

Finally, we elaborate on the rationale behind our held-out evaluation scheme. It is beneficial to emphasize that the held-out experiments are essentially evaluating prediction performance on test sets. For instance, suppose we are given a diffusion network and two different configurations of incentives. We have shown our method can predict more accurately which one will reach the activity shaping goal better. This means, in turn, that if we incentivize the users according to our method's suggestion, we will achieve the target activity better than other heuristics.

Alternatively, one can understand our evaluation scheme like this: if one applies the incentive (or intervention) levels prescribed by a method, how well the predicted outcome coincides with the reality in the test set? A good method should behavior like this: the closer the prescribed incentive (or intervention) levels to the estimated base intensities in test data, the closer the prediction based on training data to the activity level in the test data. In our experiment, the closeness in incentive level is measured by the Euclidean distance, the closeness between prediction and reality is measured by rank correlation.

## D    Temporal Properties

For the experiments on simulated objective function and held-out data we have estimated intensity from the events data. In this section, we will see how this empirical intensity resembles the theoretical intensity. We generate a synthetic network over 100 users. For each user in the generated network, we uniformly sample from $[0, 0.1]$ the exogenous intensity, and the endogenous parameters $a_{uu'}$ are uniformly sampled from $[0, 0.1]$. A bandwidth $\omega = 1$ is used in the exponential kernel. Then, the intensity is estimated empirically by dividing the number of events by the length of the respective interval.

We compute the mean and variance of the empirical activity for 100 independent runs. As illustrated in Figure 3, the average empirical intensity (the blue curve) clearly follows the theoretical instantaneous intensity (the red curve) but, as expected, as we are further from the starting point (*i.e.*, as time increases), the standard deviation of the estimates (shown in the whiskers) increases. Additionally, the green line shows the average stationary intensity. As it is expected, the instantaneous intensity tends to the stationary value when the network has been run for sufficient long time.

Figure 3: Evolution in time of empirical and theoretical intensity.

## E    Visualization of Least-squares Activity Shaping

To get a better insight on the the activity shaping problem we visualize the *least-squares activity shaping* results for the 2K and 60K datasets. Figure 4 shows the result of activity shaping at $t = 1$ targeting the same shape as in the experiments section. The red line is the target shape of the activity and the blue curve correspond to the activity profiles of users after incentivizing computed via theoretical objective. It is clear that the resulted activity behavior resembles the target shape.

(a) 2K dataset.

(b) 60K dataset.

Figure 4: Activity shaping results.

# F  Sparsity and Activity Shaping

In some applications there is a limitation on the number of users we can incentivize. In our proposed framework, we can handle this requirement by including a sparsity constraint on the optimization problem. In order to maintain the convexity of the optimization problem, we consider a $l_1$ regularization term, where a regularization parameter $\gamma$ provides the trade-off between sparsity and the activity shaping goal.

$$
\begin{aligned}
&\text{maximize}_{\boldsymbol{\mu}(t),\boldsymbol{\lambda}^{(0)}} \quad U(\boldsymbol{\mu}(t)) - \gamma||\boldsymbol{\lambda}^{(0)}||_1 \\
&\text{subject to} \quad \boldsymbol{\mu}(t) = \boldsymbol{\Psi}(t)\boldsymbol{\lambda}^{(0)}, \quad \boldsymbol{c}^\top\boldsymbol{\lambda}^{(0)} \leqslant C, \quad \boldsymbol{\lambda}^{(0)} \geqslant 0
\end{aligned}
\tag{17}
$$

Tables 2 and 3 demonstrate the effect of different values of regularization parameter on *capped activity maximization* and *minimax activity shaping*, respectively. When $\gamma$ is small, the minimum intensity is very high. On the contrary, large values of $\gamma$ imposes large penalties on the number of non-zero intensities which results in a sparse and applicable manipulation. Furthermore, this may avoid using all the budget. When dealing with unfamiliar application domains, cross validation may help to find an appropriate trade-off between sparsity and objective function.

| $\gamma$ | # Non-zeros | Budget consumed | Sum of activities |
|---|---|---|---|
| 0.5 | 2101 | 0.5 | 0.69 |
| 0.6 | 1896 | 0.46 | 0.65 |
| 0.7 | 1595 | 0.39 | 0.62 |
| 0.8 | 951 | 0.21 | 0.58 |
| 0.9 | 410 | 0.18 | 0.55 |
| 1.0 | 137 | 0.13 | 0.54 |

Table 2: Sparsity properties of capped activity maximization.

| $\gamma(\times 10^{-3})$ | # Non-zeros | Budget Consumed | $u_{min}(\times 10^{-3})$ |
|---|---|---|---|
| 0.6 | 1941 | 0.49 | 0.38 |
| 0.7 | 881 | 0.17 | 0.22 |
| 0.8 | 783 | 0.15 | 0.21 |
| 0.9 | 349 | 0.09 | 0.16 |
| 1.0 | 139 | 0.06 | 0.12 |
| 1.1 | 102 | 0.04 | 0.11 |

Table 3: Sparsity properties of minimax activity shaping.

(a) For 10,000 users.  (b) For 50,000 users.

Figure 5: Scalability of least-squares activity shaping.

(a) Capped activity maximization.  (b) Minimax activity shaping.  (c) Least-square activity shaping.

Figure 6: Activity shaping on the 60K dataset.

## G   Scalability

The naive implementation of the algorithm requires computing the matrix exponential once, and using it in (non-sparse huge) matrix-vector multiplications, *i.e.*,

$$T_{naive} = T_{\Psi} + kT_{prod}.$$

Here, $T_{\Psi}$ is the time to compute $\Psi(t)$, which itself comprised of three parts; matrix exponential computation, matrix inversion and matrix multiplications. $T_{prod}$ is the time for multiplication between the large non-sparse matrix and a vector plus the time to compute the inversion via solving linear systems of equation. Finally, $k$ is the number of gradient computations, or more generally, the number of iterations in any gradient-based iterative optimization. The dominant factor in the naive approach is the matrix exponential. It is computationally demanding and practically inefficient for more than 7000 users.

In contrast, the proposed framework benefits from the fact that the gradient depends on $\Psi(t)$ only through matrix-vector products. Thus, the running time of our activity shaping framework will be written as

$$T_{our} = kT_{grad},$$

where $T_{grad}$ is the time to compute the gradient which itself comprises the time required to solve a couple of linear systems of equations and the time to compute a couple of exponential matrix-vector multiplication.

Figure 5 demonstrates $T_{our}$ and $T_{naive}$ with respect to the number of users. For better visualization we have provided two graphs for up to 10,000 and 50,000 users, respectively. We set $k$ equal to the number of users. Since the dominant factor in the naive computation method is matrix exponential, the choice of $k$ is not that determinant. The time for computing matrix exponential is interpolated for more than 7000 users; and the interpolated total time, $T_{naive}$, is shown in red dashed line. These experiments are done in a machine equipped with one 2.5 GHz AMD Opteron Processor. This graph clearly shows the significance of designing an scalable algorithm.

Figure 6 shows the results of running our large-scale algorithm on the 60K dataset evaluated via theoretical objective function. We observe the same patterns as 2K dataset. Especially, the proposed method consistently outperforms the heuristic baselines. Heuristic baselines provide similar perfor-

mance as for the 2K dataset. DEG shows up again as a reasonable surrogate for influence, and the poor performance of XMU on activity maximization shows that high activity does not necessarily mean being more influential. For *minimax activity shaping* we observe MMASH is superior to others in $2 \times 10^{-5}$ actions per unit time, which means that the person with minimum activity uses the service $2 \times 10^{-5} \times 60 * 24 * 30 = 0.864$ times more compared to the best heuristic baseline. An increase in the activity per month of $0.864$ is not a big deal itself, however, if we consider the scale at which the network's activity is steered, we can deduce that now the service is guaranteeing, at least in theory, about $60000 \times 0.864 = 51840$ more adoptions monthly. As shown by the experiments on real-world held-out data, our approach for activity shaping outperforms all the considered heuristic baselines.

## H   Discussion

We acknowledge that our method has indeed limitations. For example, our current formulation assumes that exogenous events are constant over time. Thus, subsequent evolution in the point process is a mixture of endogenous and exogenous events. However, in practice, the shaping incentives need to be doled out throughout the evolution of the process, *e.g.*, in a sequential decision making setting. Perhaps surprisingly, our framework can be generalized to time-varying exogenous events, at the cost of stating some of the theoretical results in a convolution form, as follows:

- Lemma 1 needs to be kept in convolution form, *i.e.*, $\boldsymbol{\mu}^{(k)}(t) = \boldsymbol{G}^{(\star k)}(t) \star \boldsymbol{\lambda}^{(0)}(t)$. The sketch of the proof is very similar, and we only need to further exploit the associativity property of the convolution at the inductive step, to prove the hypothesis holds for $k + 1$:

$$\boldsymbol{\mu}^{(k+1)}(t) = \int_0^t \boldsymbol{G}(t-s) \left( \boldsymbol{G}^{(\star k)}(s) \star \boldsymbol{\lambda}^{(0)}(s) \right) ds = \boldsymbol{G}^{(\star k)}(t) \star \boldsymbol{G}(t) \star \boldsymbol{\lambda}^{(0)}(t) = \boldsymbol{G}^{(\star k+1)}(t) \star \boldsymbol{\lambda}^{(0)}(t)$$

(18)

- Lemma 2 is responsible for finding a closed form for $\widehat{\boldsymbol{G}}^{(\star k)}(z)$ and thus is not affected by a time-varying exogenous intensity. It remains unchanged.

- Theorem 3 derives the instantaneous average intensity $\boldsymbol{\mu}(t)$ and, therefore, needs to be updated accordingly using the modified Lemma 1:

$$\boldsymbol{\mu}(t) = \boldsymbol{\Psi}(t) \star \boldsymbol{\lambda}^{(0)}(t) = \left( e^{(\boldsymbol{A}-\omega\boldsymbol{I})t} + \omega(\boldsymbol{A} - \omega\boldsymbol{I})^{-1}(e^{(\boldsymbol{A}-\omega\boldsymbol{I})t} - \boldsymbol{I}) \right) \star \boldsymbol{\lambda}^{(0)}(t). \quad (19)$$

Many simple parametrized incentive functions, such as exponential incentives $\boldsymbol{\lambda}^{(0)}(t) = \boldsymbol{\lambda}^{(0)} \exp(-\alpha t)$ with constant decay $\alpha$ or constant incentives within a window $\boldsymbol{\lambda}^{(0)}(t) = \boldsymbol{\lambda}^{(0)} \boldsymbol{I}[t_1 < t < t_2]$, for a fixed window $[t_1, t_2]$, result in linear closed form expressions between the exogenous event intensity and the expected overall intensity. Nonparameteric functions result in a non-closed form expression, however, we still benefit the fact that the mapping from $\boldsymbol{\lambda}^0(t)$ to $\boldsymbol{\mu}(t)$ is linear, and hence the activity shaping problems can still be cast as convex optimization problems. In this case, the optimization can still be done via functional gradient descent (or variational calculus), though with some additional challenge to tackle.

There are many other interesting venues for future work. For example, considering competing incentives, discovering the branching structure and using it explicitly to shape the activities, exploring other possible kernel functions or even learning them using non-parametric methods remain for future work.