[Reviews · NeurIPS 2014]

Submitted by Assigned_Reviewer_8

This paper introduces the problem of activity shaping, which is a generalization of influence maximization, and allows more elaborate goal functions. The authors use multivariate Hawkes processes as the model, and via a connection to branching processes, they manage to derive a linear connection between the exogenous activity (i.e. the part that can be easily manipulated via incentives) and the overall network activity. This connection can be used in a convex optimization problem, to derive the necessary incentives to reach a global activity pattern in the network.

The paper is clearly written, it contains original research, and it is potentially a very significant contribution in the field of influence maximization.

My only comment is, that it would be nice to see the weaknesses of the proposed approach, if any.
Summary: This paper introduces a method to infer the incentives that are required for each individual to reach a global activity pattern in a network.

Submitted by Assigned_Reviewer_45

This paper deals with a problem of modeling activities in SNSs in detail, and proposes a statistical model that integrates multivariate Hawkes processes and branching Markov processes. The proposed model is feasible for explicitly modeling and estimating causality of activities based on social friendships and incentives provided by the SNS platform.

The paper is well written and motivated in general, and the proposed model is founded on a simple but reasonable theory. Although Hawkes processes have already been exploited for similar purposes, the proposed model is unique from the aspect that it provides a generic solution with convex optimization.

One drawback of this paper is the lack of qualitative analysis for both synthetic and real-world datasets, which would be more informative and interesting than quantitative evaluations for most of the readers (especially with practical minds).

Another defect lies on the experimental settings. It seems obvious that the propose method produces the best score in the settings of theoretical and simulated objectives, since the proposed method directly optimizes the score for the evaluation. I'm afraid that the held-out setting might also include a controversial issue, it might evaluate the coincidence between theoretical and simulated settings (I believe this is my misunderstanding).
Summary: The paper is well written and motivated in general, and the proposed model is founded on a simple but reasonable theory. Although the experimental part might include several potential problems, it might not be critical.

Submitted by Assigned_Reviewer_47

Quality:

The paper is technically sound, well-written, and addresses a solid problem which one could connect to practical applications. Activity shaping is at least as hard as influence maximization.

The main weakness of the paper are the experimental results. The results must necessarily be done via simulation, because activity shaping is an interventional problem, and they do not have a social network into which they could intervene. The metric for comparing heuristics is not all that compelling, but the authors take a lot of care to show their approach competes against reasonable baselines, on simulated data.

Clarity:

The paper is well written, and accessible to a reader who is not familiar with Hawkes processes. Section 2 provides a good, self-contained, explanation of the necessary background material on point processes and branching processes.

Significance:

The foundational assumption appears to be that the exogenous events (shaping) happens only at the beginning of the process. All subsequent evolution in the point process is endogenous. The paper’s assumptions diverge from reality if, for example,

1. The shaping incentives are doled out throughout the evolution of the process.
2. There are multiple competing or interacting incentives (though, I believe the Hawkes process construction could handle this case).

The authors do get something for making this assumption: "a linear relation between the expected intensity of the network and the intensity of the exogenous events" (L192), which turns activity shaping into a convex optimization.

I’d welcome some commentary on whether the Hawkes process construction would work if incentives were doled out throughout the process, e.g., in a sequential decision making setting.

Originality:

As the authors point out, Hawkes processes have had some prominence in the ML literature in the last couple of years. This paper is a solid extension that line of thinking: i.e., taking a self-exciting process, and applying it to a social network which exhibits self-exciting behavior (cascades). Solid incremental work, which describes most papers accepted at NIPS.

Once the problem is framed as a convex optimization in Section 4, the optimization itself is a straightforward projected gradient method (Algorithm 1). The main significance of the paper is the formulation of activity shaping as a convex optimization, using a Hawkes process.

Questions:

L262: Is the matrix exponential in Section 5 a consequence of the specific kernel chosen for this paper, or is it required for any kernel G ?

Misc:

Bibliography: Check your Bibtex (e.g., “hawkes” vs. “Hawkes”)
Summary: Introduces the activity shaping problem, and proposes a novel approach to it, quite different from what I’ve seen in the influence maximization literature.
Author Feedback
Author rebuttal: Thank you for your careful reading and detailed comments; these will help improve the final version of our paper if accepted.

Reviewer 45:

** Qualitative Results **
We agree that a qualitative analysis of both synthetic and real-world dataset may be very valuable for an interested
reader. Due to space constraints, we have reported a few key qualitative results in our paper:
- In capped activity maximization (CAM): Surprisingly, the most influential users are not necessarily the most active ones. In contrast, degree is a good approximation of the influence, but our method performs better than degree based heuristic.
- In minimax activity shaping (MMASH): Peer influence is an important factor for encouraging inactive users.
Directly trying to incentivize inactive user will not work (the MINMU competitor). Furthermore, our method differs from LP-based heuristic only in the way of using peer influence and performs much better in this task.

We also note that our results have practical implications for real twitter data (discussed in the paper):
- In capped activity maximization (CAM): our method results in 34,560 more events than the best competitor per month in a network of 2441 users. Moreover, in held-out data, it can predict the ordering of the activity shaping goal correctly for 122 more pairs of incentivizations compared to the second best.
- In minimax activity shaping (MMASH): our method prompts the least active user to take 4.3 more actions compared to the best competitor over a month in 2K dataset and 0.86 more actions in 60K dataset.

Due to page limit, many other qualitative results are also provided in the appendix:
- Appendix D contains an illustration of the behavior of least squares activity shaping (LSASH). The singular peaks on the activity profile are probably the most effective users for the sake of reaching to the red target. It can been seen that some users are receiving much more incentives than others since they can steer the network to the desired shapes.
- Appendix E contains an illustrative example of the effect of sparsity on activity shaping. Sparsity limits the number of users we can incentivize and decrease the effectiveness of achieving the activity shaping goals.
- In all three tasks, longer times lead to larger differences between our method and the alternatives. This occurs because the longer the time, the more endogenous activity is triggered by network influence, and thus our framework, which models both endogenous and exogenous events, becomes more suitable.

We will describe more clearly the qualitative results in the final version of the paper, highlighting their practical
implications. Moreover, we will perform a more detailed qualitative analysis in a longer journal version
of our work.

** Held-out Experiments **

We emphasize that the held-out experiments are essentially evaluating prediction performance on test sets. For instance, suppose we are given a diffusion network and two different configuration of incentives. Our method can predict more accurately which one will reach the activity shaping goal better. This means, in turn, that if we incentivize the users according to our method's suggestion, we will achieve the target activity better than other heuristics.

Alternatively, one can understand our evaluation scheme like this: if one applies the incentive (or intervention) levels prescribed by a method, how well the predicted outcome coincides with the reality in the test set? A good method should behavior like this: the closer the prescribed incentive (or intervention) levels to the estimated base intensities in test data, the closer the prediction based on training data to the activity level in the test data. In our experiment, the closeness in incentive level is measured by the Euclidean distance, the closeness between prediction and reality is measured by rank correlation.

Reviewer 47:

Since we do not have control over a real social platform, we design the held-out experiments to closely mimic real intervention scenario. (Please see also the Held-Out Experiment section for reviewer 45).

We note that our current formulation assumes that exogeneous events (shaping) are constant over time. Thus subsequent evolution in the point process is a mixture of endogenous and exogeneous events. In fact, our framework can be generalized to time-varying exogeneous events, but the theories need to be stated in a convolution form:
- Lemma 1 needs to be kept in convolution form. Many simple parametrized functions, such as decaying incentives \lambda^0(t)=\lambda^0 * exp(-a*t) with scaler constant a, constant incentives within a window \lambda^0(t) = \lambda^0 * I[t1 < t < t2] for a fixed window [t1, t2], will result in linear closed form.
If we're seeking a nonparameteric function we benefit the fact that the mapping from lambda^0(t) to mu(t) is linear in this general case, and hence will still result in convex optimization later. In principle, the optimization can still be done via functional gradient descent (or variational calculus), though with some additional challenge to tackle.
- Lemma 2 will not change.
- Theorem 3: needs to be updated accordingly as a consequence of the change Lemma 1.

We emphasize that the linear relation between the exogenous and expected intensity is *not* an assumption but an original result derived in the paper.

Reviewer 8:

We agree with the reviewer that it would be interesting to point out some of the weaknesses of our method. We acknowledge that our method has indeed limitations. For example, it does not consider competing incentives or time-varying mutual excitation, which are interesting future research directions. We will include more discussion on this in the final version of our paper.